# Deep State Space Models for Unconditional Word Generation

**Florian Schmidt**
Department of Computer Science
ETH Zürich
florian.schmidt@inf.ethz.ch

**Thomas Hofmann**
Department of Computer Science
ETH Zürich
thomas.hofmann@inf.ethz.ch

## Abstract

Autoregressive feedback is considered a necessity for successful unconditional text generation using stochastic sequence models. However, such feedback is known to introduce systematic biases into the training process and it obscures a principle of generation: committing to global information and forgetting local nuances. We show that a non-autoregressive deep state space model with a clear separation of global and local uncertainty can be built from only two ingredients: An independent noise source and a deterministic transition function. Recent advances on flow-based variational inference can be used to train an evidence lower-bound without resorting to annealing, auxiliary losses or similar measures. The result is a highly interpretable generative model on par with comparable auto-regressive models on the task of word generation.

## 1 Introduction

Deep generative models for sequential data are an active field of research. Generation of text, in particular, remains a challenging and relevant area [HYX+17]. Recurrent neural networks (RNNs) are a common model class, and are typically trained via maximum likelihood [BVV+15] or adversarially [YZWY16, FGD18]. For conditional text generation, the sequence-to-sequence architecture of [SVL14] has proven to be an excellent starting point, leading to significant improvements across a range of tasks, including machine translation [BCB14, VSP+17], text summarization [RCW15], sentence compression [FAC+15] and dialogue systems [SSB+16]. Similarly, RNN language models have been used with success in speech recognition [MKB+10, GJ14]. In all these tasks, generation is conditioned on information that severely narrows down the set of likely sequences. The role of the model is then largely to distribute probability mass within relatively constrained sets of candidates.

Our interest is, by contrast, in unconditional or free generation of text via RNNs. We take as point of departure the shortcomings of existing model architectures and training methodologies developed for conditional tasks. These arise from the increased challenges on both, accuracy and coverage. Generating grammatical and coherent text is considerably more difficult without reliance on an acoustic signal or a source sentence, which may constrain, if not determine much of the sentence structure. Moreover, failure to sufficiently capture the variety and variability of data may not surface in conditional tasks, yet is a key desideratum in unconditional text generation.

The *de facto* standard model for text generation is based on the RNN architecture originally proposed by [Gra13] and incorporated as a decoder network in [SVL14]. It evolves a continuous state vector, emitting one symbol at a time, which is then fed back into the state evolution – a property that characterizes the broader class of autoregressive models. However, even in a conditional setting, these RNNs are difficult to train without substitution of previously generated words by ground truth observations during training, a technique generally referred to as *teacher forcing* [WZ89]. This approach is known to cause biases [RCAZ15a, GLZ+16] that can be detrimental to test time

performance, where such nudging is not available and where state trajectories can go astray, requiring *ad hoc* fixes like beam search [WR16] or scheduled sampling [BVJS15]. Nevertheless, teacher forcing has been carried over to unconditional generation [BVV+15].

Another drawback of autoregressive feedback [Gra13] is in the dual use of a single source of stochasticity. The probabilistic output selection has to account for the local variability in the next token distribution. In addition, it also has to inject a sufficient amount of entropy into the evolution of the state space sequence, which is otherwise deterministic. Such noise injection is known to compete with the explanatory power of autoregressive feedback mechanisms and may result in degenerate, near deterministic models [BVV+15]. As a consequence, there have been a variety of papers that propose deep stochastic state sequence models, which combine stochastic and deterministic dependencies, e.g. [CKD+15, FSPW16], or which make use of auxiliary latent variables [GSC+17], auxiliary losses [SATB17], and annealing schedules [BVV+15]. No canonical architecture has emerged so far and it remains unclear how the stochasticity in these models can be interpreted and measured.

In this paper, we propose a stochastic sequence model that preserves the Markov structure of standard state space models by cleanly separating the stochasticity in the state evolution, injected via a white noise process, from the randomness in the local token generation. We train our model using variational inference (VI) and build upon recent advances in normalizing flows [RM15, KSW16] to define rich enough stochastic state transition functions for both, generation and inference. Our main goal is to investigate the fundamental question of how far one can push such an approach in text generation, and to more deeply understand the role of stochasticity. For that reason, we have used the most basic problem of text generation as our testbed: word morphology, i.e. the mechanisms underlying the formation of words from characters. This enables us to empirically compare our model to autoregressive RNNs on several metrics that are intractable in more complex tasks such as word sequence modeling.

## 2 Model

We argue that text generation is subject to two sorts of uncertainty: Uncertainty about plausible long-term continuations and uncertainty about the emission of the current token. The first reflects the entropy of all things considered "natural language", the second reflects symbolic entropy at a fixed position that arises from ambiguity, (near-)analogies, or a lack of contextual constraints. As a consequence, we cast the emission of a token as a fundamental trade-off between *committing* and *forgetting* about information.

### 2.1 State space model

Let us define a state space model with *transition function*

$$F : \mathbb{R}^d \times \mathbb{R}^d \to \mathbb{R}^d, \quad (\mathbf{h}_t, \boldsymbol{\xi}_t) \mapsto \mathbf{h}_{t+1} = F(\mathbf{h}_t, \boldsymbol{\xi}_t), \quad \boldsymbol{\xi}_t \overset{\text{iid}}{\sim} \mathcal{N}(\mathbf{0}, \mathbf{I}). \tag{1}$$

$F$ is deterministic, yet driven by a white noise process $\boldsymbol{\xi}$, and, starting from some $\mathbf{h}_0$, defines a homogeneous stochastic process. A local observation model $P(w_t|\mathbf{h}_t)$ generates symbols $w_t \in \Sigma$ and is typically realized by a softmax layer with symbol embeddings.

The marginal probability of a symbol sequence $\mathbf{w} = w_{1:T}$ is obtained by integrating out $\mathbf{h} = \mathbf{h}_{1:T}$,

$$P(\mathbf{w}) = \int \prod_{t=1}^{T} p(\mathbf{h}_t|\mathbf{h}_{t-1}) P(w_t|\mathbf{h}_t) \, d\mathbf{h}. \tag{2}$$

Here $p(\mathbf{h}_t|\mathbf{h}_{t-1})$ is defined implicitly by driving $F$ with noise as we will explain in more detail below.[1] In contrast to common RNN architectures, we have defined $F$ to *not* include an auto-regressive input, such as $w_{t-1}$, making potential biases as in teacher-forcing a non-issue. Furthermore, this implements our assumption about the role of entropy and information for generation. The information about the local outcome under $P(w_t|\mathbf{h}_t)$ is not considered in the transition to the next state as there is no feedback. Thus in this model, all entropy about possible sequence continuations *must* arise from the noise process $\boldsymbol{\xi}$, which cannot be ignored in a successfully trained model.

The implied generative procedure follows directly from the chain rule. To sample a sequence of observations we (i) sample a white noise sequence $\boldsymbol{\xi} = \boldsymbol{\xi}_{1...T}$ (ii) deterministically compute $\mathbf{h} = \mathbf{h}_{1...T}$ from $\mathbf{h}_0$ and $\boldsymbol{\xi}$ via $F$ and (iii) sample from the observation model $\prod_{t=1}^{T} P(w_t|\mathbf{h}_t)$. The remainder of this section focuses on how we can define a sufficiently powerful familiy of state evolution functions $F$ and how variational inference can be used for training.

## 2.2 Variational inference

Model-based variational inference (VI) allows us to approximate the marginalization in Eq. (2) by posterior expectations with regard to an inference model $q(\mathbf{h}|\mathbf{w})$. It is easy to verify that the true posterior obeys the conditional independences $\mathbf{h}_t \perp\!\!\!\perp \text{rest} \,|\, \mathbf{h}_{t-1}, \mathbf{w}_{t:T}$, which informs our design of the inference model, cf. [FSPW16]:

$$q(\mathbf{h}|\mathbf{w}) = \prod_{t=1}^{T} q(\mathbf{h}_t|\mathbf{h}_{t-1}, \mathbf{w}_{t:T}) \,. \tag{3}$$

This is to say, the previous state is a sufficient summary of the past. Jensen's inequality then directly implies the evidence lower bound (ELBO)

$$\log P(\mathbf{w}) \geq \mathbb{E}_q \left[ \log P(\mathbf{w}|\mathbf{h}) + \log \frac{p(\mathbf{h})}{q(\mathbf{h}|\mathbf{w})} \right] =: \mathcal{L} = \sum_{t=1}^{T} \mathcal{L}_t \tag{4}$$

$$\mathcal{L}_t := \mathbb{E}_q \left[ \log P(\mathbf{w}_t|\mathbf{h}_t) \right] + \mathbb{E}_q \left[ \log \frac{p(\mathbf{h}_t|\mathbf{h}_{t-1})}{q(\mathbf{h}_t|\mathbf{h}_{t-1}, \mathbf{w}_{t:T})} \right] \tag{5}$$

This is a well-known form, which highlights the per-step balance between prediction quality and the discrepancy between the transition probabilities of the unconditioned generative and the data-conditioned inference models [FSnPW16, CKD+15]. Intuitively, the inference model breaks down the long range dependencies and provides a local training signal to the generative model for a single step transition and a single output generation.

Using VI successfully for generating symbol sequences requires parametrizing powerful yet tractable next state transitions. As a minimum requirement, forward sampling and log-likelihood computation need to be available. Extensions of VAEs [RM15, KSW16] have shown that for non-sequential models under certain conditions an invertible function $\mathbf{h} = f(\boldsymbol{\xi})$ can shape moderately complex distributions over $\boldsymbol{\xi}$ into highly complex ones over $\mathbf{h}$, while still providing the operations necessary for efficient VI. The authors show that a bound similar to Eq. (5) can be obtained by using the law of the unconscious statistician [RM15] and a density transformation to express the discrepancy between generative and inference model in terms of $\boldsymbol{\xi}$ instead of $\mathbf{h}$

$$\mathcal{L} = \mathbb{E}_{q(\boldsymbol{\xi}|\mathbf{w})} \left[ \log P(\mathbf{w}|f(\boldsymbol{\xi})) + \log \frac{p(f(\boldsymbol{\xi}))}{q(\boldsymbol{\xi}|\mathbf{w})} + \log |\det \mathbf{J}_f(\boldsymbol{\xi})| \right] \tag{6}$$

This allows the inference model to work with an implicit latent distribution at the price of computing the Jacobian determinant of $f$. Luckily, there are many choices such that this can be done in $\mathcal{O}(d)$ [RM15, DSB16].

## 2.3 Training through coupled transition functions

We propose to use two separate transition functions $F_q$ and $F_g$ for the inference and the generative model, respectively. Using results from flow-based VAEs we derive an ELBO that reveals the intrinsic coupling of both and expresses the relation of the two as a part of the objective that is determined solely by the data. A shared transition model $F_q = F_g$ constitutes a special case.

**Two-Flow ELBO** For a transition function $F$ as in Eq. (1) fix $\mathbf{h} = \mathbf{h}_*$ and define the restriction $f(\boldsymbol{\xi}) = F(\mathbf{h}, \boldsymbol{\xi})_{|\mathbf{h}=\mathbf{h}_*}$. We require that for any $\mathbf{h}_*$, $f$ is a diffeomorphism and thus has a differentiable inverse. In fact, as we work with (possibly) different $F_g$ and $F_q$ for generation and inference, we have restrictions $f_g$ and $f_q$, respectively. For better readability we will omit the conditioning variable $\mathbf{h}_*$ in the sequel.

By combining the per-step decomposition in (5) with the flow-based ELBO from (6), we get (implicitly setting $\mathbf{h}_* = \mathbf{h}_{t-1}$):

$$\mathcal{L}_t = \mathbb{E}_{q(\boldsymbol{\xi}|\mathbf{w})} \left[ \log P(w_t|f_q(\boldsymbol{\xi}_t)) + \log \frac{p(f_q(\boldsymbol{\xi}_t)|\mathbf{h}_{t-1})}{q(\boldsymbol{\xi}_t|\mathbf{h}_{t-1}; w_{t:T})} + \log \left| \det \mathbf{J}_{f_q}(\boldsymbol{\xi}_t) \right| \right] . \qquad (7)$$

As our generative model also uses a flow to transform $\boldsymbol{\xi}_t$ into a distribution on $\mathbf{h}_t$, it is more natural to use the (simple) density in $\boldsymbol{\xi}$-space. Performing another change of variable, this time on the density of the generative model, we get

$$p(\mathbf{h}_t|\mathbf{h}_{t-1}) = p(\boldsymbol{\zeta}_t|\mathbf{h}_{t-1}) \cdot |\det \mathbf{J}_{f_g^{-1}}(f_q(\boldsymbol{\xi}_t))| = \frac{r(\boldsymbol{\zeta}_t)}{|\det \mathbf{J}_{f_g}(\boldsymbol{\zeta}_t)|}, \quad \boldsymbol{\zeta}_t := (f_g^{-1} \circ f_q)(\boldsymbol{\xi}_t) \qquad (8)$$

where $r$ now is simply the (multivariate) standard normal density as $\boldsymbol{\xi}_t$ does not depend $\mathbf{h}_{t-1}$, whereas $\mathbf{h}_t$ does. We have introduced new noise variable $\boldsymbol{\zeta}_t = s(\boldsymbol{\xi}_t)$ to highlight the importance of the transformation $s = f_g^{-1} \circ f_q$, which is a combined flow of the forward inference flow and the inverse generative flow. Essentially, it follows the suggested $\boldsymbol{\xi}$-distribution of the inference model into the latent state space and back into the noise space of the generative model with its uninformative distribution. Putting this back into Eq. (7) and exploiting the fact that the Jacobians can be combined via $\det \mathbf{J}_s = \det \mathbf{J}_{f_q}/\det \mathbf{J}_{f_g}$ we finally get

$$\mathcal{L}_t = \mathbb{E}_{q(\boldsymbol{\xi}|\mathbf{w})} \left[ \log P(w_t|f_q(\boldsymbol{\xi}_t)) + \log \frac{r(s(\boldsymbol{\xi}_t))}{q(\boldsymbol{\xi}_t|\mathbf{h}_{t-1}; w_{t:T})} + \log |\det \mathbf{J}_s(\boldsymbol{\xi}_t)| \right] . \qquad (9)$$

**Interpretation** Naïvely employing the model-based ELBO approach, one has to learn two independently parametrized transition models $p(\mathbf{h}_t|\mathbf{h}_{t-1})$ and $q(\mathbf{h}_t|\mathbf{h}_{t-1}, w_{t...T})$, one informed about the future and one not. Matching the two then becomes and integral part of the objective. However, since the transition model encapsulates most of the model complexity, this introduces redundancy where the learning problem is most challenging. Nevertheless, generative and inference model do address the transition problem from very different angles. Therefore, forcing both to use the exact same transition model might limit flexibility during training and result in an inferior generative model. Thus our model casts $F_g$ and $F_q$ as independently parametrized functions that are coupled through the objective by treating them as proper transformations of an underlying white noise process. [2]

**Special cases** Additive Gaussian noise $\mathbf{h}_{t+1} = \mathbf{h}_t + \boldsymbol{\xi}_t$ can be seen as the simplest form of $F_g$ or, alternatively, as a generative model without flow (as $\mathbf{J}_{f_g} = \mathbf{I}$). Of course, repeated addition of noise does not provide a meaningful latent trajectory. Finally, note that for $F_g = F_q$, $s = \text{id}$ and the nominator in the second term becomes a simple prior probability $r(\boldsymbol{\xi}_t)$, whereas the determinant reduces to a constant. We now explore possible candidates for the flows in $F_g$ and $F_q$.

### 2.4 Families of transition functions

Since the Jacobian of a composed function factorizes, a flow $F$ is often composed of a chain of individual invertible functions $F = F_k \circ \cdots \circ F_1$ [RM15]. We experiment with individual functions

$$F(\mathbf{h}_{t-1}, \boldsymbol{\xi}_t) = g(\mathbf{h}_{t-1}) + \mathbf{G}(\mathbf{h}_{t-1})\boldsymbol{\xi}_t \qquad (10)$$

where $g$ is a multilayer MLP $\mathbb{R}^d \to \mathbb{R}^d$ and $\mathbf{G}$ is a neural network $\mathbb{R}^d \to \mathbb{R}^d \times \mathbb{R}^d$ mapping $\mathbf{h}_{t-1}$ to a lower-triangular $d \times d$ matrix with non-zero diagonal entries. Again, we use MLPs for this mapping and clip the diagonal away from $[-\delta, \delta]$ for some hyper parameter $0 < \delta < 0.5$. The lower-triangular structure allows computing the determinant in $\mathcal{O}(d)$ and stable inversion of the mapping by substitution in $\mathcal{O}(d^2)$. As a special case we also consider the case when $\mathbf{G}$ is restricted to diagonal matrices. Finally, we experiment with a conditional variant of the Real NVP flow [DSB16].

Computing $F_g^{-1}$ is central to our objective and we found that depending on the flow actually parametrizing the inverse directly results in more stable and efficient training.

## 2.5 Inference network

So far we have only motivated the factorization of the inference network $q(\mathbf{h}|w) = \prod q(\mathbf{h}_t|\mathbf{h}_{t-1}, w_{t:T})$ but treated it as a black-box otherwise. Remember that sampling from the inference network amounts to sampling $\xi_t \sim q(\cdot|\mathbf{h}_{t-1}, w_{t...T})$ and then performing the deterministic transition $F_q(\mathbf{h}_{t-1}, \xi_t)$. We observe much better training stability when conditioning $q$ on the data $w_{t...T}$ only and modeling interaction with $\mathbf{h}_{t-1}$ exclusively through $F_q$. This coincides with our intuition that the two inputs to a transition function provide semantically orthogonal contributions.

We follow existing work [DSB16] and choose $q$ as the density of a normal distribution with diagonal covariance matrix. We follow the idea of [FSPW16] and incorporate the variable-length sequence $w_{t:T}$ by conditioning on the state of an RNN running backwards in time across $w_{1...T}$. We embed the symbols $w_{1...T}$ in a vector space $\mathbb{R}^{d_E}$ and use use a GRU cell to produce a sequence of hidden states $\mathbf{a}_T, \ldots, \mathbf{a}_1$ where $\mathbf{a}_t$ has digested tokens $w_{t:T}$. Together $\mathbf{h}_{t-1}$ and $\mathbf{a}_t$ parametrize the mean and co-variance matrix of $q$.

## 2.6 Optimization

Except in very specific and simple cases, for instance, a Kalman filter, it will not be possible to efficiently compute the $q$-expectations in Eq. (5) exactly. Instead, we sample $q$ in every time-step as is common practice for sequential ELBOs [FSnPW16, GSC$^+$17]. The re-parametrization trick allows pushing all necessary gradients through these expectations to optimize the bound via stochastic gradient-based optimization techniques such as Adam [KB14].

## 2.7 Extension: Importance-weighted ELBO for tracking the generative model

Conceptionally, there are two ways we can imagine an inference network to propose $\boldsymbol{\xi}_{1:T}$ sequences for a given sentence $w_{1:T}$. Either, as described above, by digesting $w_{1...T}$ right-to-left and proposing $\boldsymbol{\xi}_{1:T}$ left-to-right. Or, by iteratively proposing a $\boldsymbol{\xi}_t$ taking into account the last state $\mathbf{h}_{t-1}$ proposed *and* the generative deterministic mechanism $F_g$. The latter allows the inference network to peek at states $\mathbf{h}_t$ that $F_g$ *could* generate from $\mathbf{h}_{t-1}$ before proposing an actual target $\mathbf{h}_t$. This allows the inference model to track a multi-modal $F_g$ without need for $F_q$ to match its expressiveness. As a consequence, this might offer the possibility to learn multi-modal generative models, without the need to employ complex multi-modal distributions in the inference model.

Our extension is built on importance weighted auto-encoders (IWAE) [BGS15]. The IWAE ELBO is derived by writing the log marginal as a Monte Carlo estimate *before* using Jensen's inequality. The result is an ELBO and corresponding gradients of the form[3]

$$\mathcal{L} = \mathbb{E}_{\mathbf{h}^{(k)}}\left[\log \frac{1}{K}\sum_{k=1}^{K}\underbrace{\frac{p(\mathbf{w}, \mathbf{h}^{(k)})}{q(\mathbf{h}^{(k)}|\mathbf{w})}}_{=:\,\omega^{(k)}}\right], \quad \nabla\mathcal{L} = \mathbb{E}_{\mathbf{h}^{(k)}}\left[\sum_{k=1}^{K}\frac{\omega^{(k)}}{\sum_{k'}\omega^{(k')}}\nabla\log\omega^{(k)}\right], \quad \mathbf{h}^{(k)}\sim q(\cdot|\mathbf{w}) \quad (11)$$

The authors motivate (11) as a weighting mechanism relieving the inference model from explaining the data well with *every* sample. We will use the symmetry of this argument to let the inference model condition on potential next states $\mathbf{h}_{g_t} = F_g(\mathbf{h}_{t-1}, \boldsymbol{\xi}_t), \boldsymbol{\xi}_t \sim \mathcal{N}(0, I)$ from the generative model without requiring *every* $\mathbf{h}_{g_t}$ to allow $q$ to make a good proposal. In other words, the $K$ sampled outputs of $F_g$ become a vectorized representation of $F_g$ to condition on. In our sequential model, computing $\omega^{(k)}$ exactly is intractable as it would require rolling out the network until time $T$. Instead, we limit the horizon to only one time-step. Although this biases the estimate of the weights and consequently the ELBO, longer horizons did empirically not show benefits. When proceeding to time-step $t + 1$ we choose the new hidden state by sampling $\mathbf{h}^{(k)}$ with probability proportionally to $\omega^{(k)}$. Algorithm 1 summarizes the steps carried out at time $t$ for a given $\mathbf{h}_{t-1}$ (to not overload the notation, we drop $t$ in $\mathbf{h}_{g_t}$) and a more detailed derivation of the bound is given in Appendix A.

**Algorithm 1** Detailed forward pass with importance weighting

---

Simulate $F_g$:                 $\mathbf{h}_g^{(k)} = F_g(\mathbf{h}_{t-1}, \boldsymbol{\xi}^{(k)})$, where $\boldsymbol{\xi}^{(k)} \sim \mathcal{N}(0, I)$, $k = 1, \ldots, K$

Instantiate the inference family:    $q_k(\mathbf{h}) = q(\mathbf{h}|\mathbf{h}_g^{(k)}, \mathbf{h}_{t-1}, w_{t:T})$

Sample inference:             $\mathbf{h}^{(k)} \sim q_k$

Compute gradients as in (11) where $\omega^{(k)} = P(w_t|\mathbf{h}^{(k)})p(\mathbf{h}^{(k)}|\mathbf{h}_{t-1})/q_k(\mathbf{h}^{(k)})$

Sample $\mathbf{h}^{(k)}$ according to $\omega^{(1)} \ldots \omega^{(K)}$ for the next step.

---

## 3 Related Work

Our work intersects with work directly addressing teacher-forcing, mostly on language modelling and translation (which are mostly not state space models) and stochastic state space models (which are typically autoregressive and do not address teacher forcing).

Early work on addressing teacher-forcing has focused on mitigating its biases by adapting the RNN training procedure to partly rely on the model's prediction during training [BVJS15, RCAZ15b]. Recently, the problem has been addressed for conditional generation within an adversarial framework [GLZ+16] and in various learning to search frameworks [WR16, LAOL17]. However, by design these models do not perform stochastic state transitions.

There have been proposals for hybrid architectures that augment the deterministic RNN state sequences by chains of random variables [CKD+15, FSPW16]. However, these approaches are largely patching-up the output feedback mechanism to allow for better modeling of local correlations, leaving the deterministic skeleton of the RNN state sequence untouched. A recent evolution of deep stochastic sequence models has developed models of ever increasing complexity including intertwined stochastic and deterministic state sequences [CKD+15, FSPW16] additional auxiliary latent variables [GSC+17] auxiliary losses [SATB17] and annealing schedules [BVV+15]. At the same time, it remains often unclear how the stochasticity in these models can be interpreted and measured.

Closest in spirit to our transition functions is work by Maximilian et al.[KSBvdS17] on generation with external control inputs. In contrast to us they use a simple mixture of linear transition functions and work around using density transformations akin to [BO14]. In our unconditional regime we found that relating the stochasticity in $\boldsymbol{\xi}$ explicitly to the stochasticity in $\mathbf{h}$ is key to successful training. Finally, variational conditioning mechanisms similar in spirit to ours have seen great success in image generation[GDGW15].

Among generative unconditional sequential models GANs are as of today the most prominent architecture [YZWY16, JKMHL16, FGD18, CLZ+17]. To the best of our knowledge, our model is the first non-autoregressive model for sequence generation in a maximum likelihood framework.

## 4 Evaluation

Naturally, the quality of a generative model must be measured in terms of the quality of its outputs. However, we also put special emphasis on investigating whether the stochasticity inherent in our model operates as advertised.

### 4.1 Data Inspection

Evaluating generative models of text is a field of ongoing research and currently used methods range from simple data-space statistics to expensive human evaluation [FGD18]. We argue that for morphology, and in particular non-autoregressive models, there is an interesting middle ground: Compared to the space of all sentences, the space of all words has still moderate cardinality which allows us to estimate the data distribution by unigram word-frequencies. As a consequence, we can reliably approximate the cross-entropy which naturally generalizes data-space metrics to probabilistic models and addresses both, over-generalization (assigning non-zero probability to non-existing words) and over-confidence (distributing high probability mass only among a few words).

This metric can be addressed by all models which operate by first stochastically generating a sequence of hidden states and then defining a distribution over the data-space given the state sequence. For our

model we approximate the marginal by a Monte Carlo estimate of (2)

$$P(\mathbf{w}) = \int P(\mathbf{w}|\mathbf{h})p(\mathbf{h})d\mathbf{h} = \frac{1}{K}\sum_{k=1}^{K} P(\mathbf{w}|\mathbf{h}^{(k)}), \quad \mathbf{h}^{(k)} \sim p(\mathbf{h}) \tag{12}$$

Note that sampling from $p(\mathbf{h})$ boils down to sampling $\xi_{1...T}$ from independent standard normals and then applying $F_g$. In particular, the non-autoregressive property of our model allows us to estimate all words in some set $S$ using $K$ samples each by using only $K$ independent trajectories $\mathbf{h}$ overall.

Finally, we include two data-space metrics as an intuitive, yet less accurate measure. From a collection of generated words, we estimate (i) the fraction of words that are in the training vocabulary ($\mathbf{w} \in V$) and (ii) the fraction of *unique* words that are in the training vocabulary ($\mathbf{w} \in V$ unique).[4]

### 4.2 Entropy Inspection

We want to go beyond the usual evaluation of existing work on stochastic sequence models and also assess the quality of our noise model. In particular, we are interested in how much information contained in a state $\mathbf{h}_t$ about the output $P(w_t|\mathbf{h}_t)$ is due to the corresponding noise vector $\boldsymbol{\xi}_t$. This is quantified by the mutual information between the noise $\xi_t$ and the observation $w_t$ given the noise $\xi_{1:t-1}$ that defined the prefix up to time $t$. Since $\mathbf{h}_{t-1}$ is a deterministic function of $\xi_{1:t-1}$, we write

$$I(t) = I(w_t; \xi_t|\mathbf{h}_{t-1}) = \mathbb{E}_{\mathbf{h}_{t-1}}\left[H[w_t|\mathbf{h}_{t-1}] - H[w_t|\xi_t, \mathbf{h}_{t-1}]\right] \geq 0 \tag{13}$$

to quantify the dependence between noise and observation at one time-step. For a model ignoring the noise variables, knowledge of $\xi_t$ does not reduce the uncertainty about $w_t$, so that $I(t) = 0$. We can use Monte Carlo estimates for all expectations in (13).

## 5 Experiments

### 5.1 Dataset and baseline

For our experiments, we use the BooksCorpus [KZS$^+$15, ZKZ$^+$15], a freely available collection of novels comprising of almost 1B tokens out of which 1.3M are unique. To filter out artefacts and some very uncommon words found in fiction, we restrict the vocabulary to words of length $2 \leq l \leq 12$ with at least 10 occurrences that only contain letters resulting in a 143K vocabulary. Besides the standard 10% test-train split at the word level, we also perform a second, alternative split at the vocabulary level. That means, 10 percent of the words, chosen *regardless* of their frequency, will be unique to the test set. This is motivated by the fact that even a small test-set under the former regime will result in only very few, very unlikely words unique to the test-set. However, generalization to unseen words is the essence of morphology. As an additional metric to measuring generalization in this scenario, we evaluate the generated output under Witten-Bell discounted character $n$-gram models trained on either the whole corpus or the test data only.

Our baseline is a GRU cell and the standard RNN training procedure with teacher-forcing[5]. Hidden state size and embedding size are identical to our model's.

### 5.2 Model parametrization

We stick to a standard softmax observation model and instead focus the model design on different combinations of flows for $F_g$ and $F_q$. We investigate the flow in Equation (10), denoted as TRIL, its diagonal version DIAG and a simple identity ID. We denote repeated application of (independently parametrized) flows as in $2 \times$ TRIL. For the weighted version we use $K \in \{2, 5, 10\}$ samples. In addition, for $F_g$ we experiment with a sequence of Real NVPs with masking dimensions $d = 2 \ldots 7$ (two internal hidden layers of size 8 each). Furthermore, we investigate deviating from the factorization (3) by using a bidirectional RNN conditioning on all $w_{1...T}$ in every timestep. Finally, for the best performing configuration, we also investigate state-sizes $d = \{16, 32\}$.

## 5.3 Results

Table 1 shows the result for the standard split. By $\pm$ we indicate mean and standard deviation across 5 or 10 (for IWAE) identical runs[6]. The data-space metrics require manually trading off precision and coverage. We observe that two layers of the TRIL flow improve performance. Furthermore, importance weighting significantly improves the results across all metrics with diminishing returns at $K = 10$. Its effectiveness is also confirmed by an increase in variance across the weights $\omega^1 \ldots \omega^T$ during training which can be attributed to the significance of the noise model (see 5.4 for more details). We found training with REAL-NVP to be very unstable. We attribute the relatively poor performance of NVP to the sequential VI setting which deviates heavily from what it was designed for and keep adaptions for future work.

| Model | $H[P_{\text{train}}, \hat{P}]$ | $H[P_{\text{test}}, \hat{P}]$ | $\mathbf{w} \in V$ unique | $\mathbf{w} \in V$ | $\bar{I}$ |
|---|---|---|---|---|---|
| TRIL | 12.13±.11 | 11.99±.11 | 0.18±.00 | 0.43±.03 | 0.95±.04 |
| TRIL, K=2 | 11.76±.12 | 11.82±.12 | 0.16±.01 | 0.46±.02 | 1.06±.16 |
| TRIL, K=5 | 11.46±.05 | 11.51±.05 | 0.16±.01 | 0.48±.02 | 1.08±.13 |
| TRIL, K=10 | 11.43±.05 | 11.47±.05 | 0.16±.01 | 0.49±.02 | 1.12±.12 |
| 2×TRIL | 11.91±.08 | 11.86±.13 | 0.17±.01 | 0.45±.02 | 0.89±.07 |
| 2×TRIL, K=2 | 11.55±.09 | 11.61±.09 | 0.16±.00 | 0.47±.01 | 1.00±.13 |
| 2×TRIL, K=5 | 11.42±.07 | 11.46±.06 | 0.16±.00 | 0.49±.01 | 1.20±.12 |
| 2×TRIL, K=10 | **11.33±.05** | **11.38±.06** | 0.16±.00 | 0.49±.01 | **1.28±.13** |
| 2×TRIL, K=10, BIDI | **11.33±.09** | 11.39±.10 | 0.16±.01 | 0.48±.00 | 1.25±.16 |
| $d = 16$ 2×TRIL, K=10 | 11.21 | 11.43 | 0.15 | 0.48 | 1.43 |
| $d = 32$ 2×TRIL, K=10 | 11.27 | 11.13 | 0.15 | 0.50 | 1.31 |
| REAL-NVP-[2,3,4,5,6,7] | 11.77 | 11.81 | 0.12 | 0.53 | 0.94 |
| BASELINE-8D | 12.92 | 12.97 | 0.13 | 0.53 | – |
| BASELINE-16D | 12.55 | 12.60 | 0.14 | 0.62 | – |
| ORACLE-TRAIN | 7.0 | 7.02[7] | 0.27 | 1.0 | – |

Table 1: Results on generation. The cross entropy is computed wrt. both training and test set. ORACLE-TRAIN is a model sampling from the training data.

Interestingly, our standard inference model is on par with the equivalently parametrized bidirectional inference model suggesting that historic information can be sufficiently stored in the states and confirming d-separation as the right principle for inference design.

The poor cross-entropy achieved by the baseline can partly be explained by the fact that autoregressive RNNs are trained on conditional next-word-predictions. Estimating the real data-space distribution would require aggregating over all *possible* sequences $\mathbf{w} \in V^T$. However, the data-space metrics clearly show that the performance cannot solely be attributed to this.

Table 2 shows that generalization for the alternative split is indeed harder but cross entropy results carry over from the standard setting. Here we sample trajectories and extract the argmax from the observation model which resembles more closely the procedure of the baseline. Under $n$-gram perplexity both models are on par with a slight advantage of the baseline on longer $n$-grams and slightly better generalization of our proposed model.

| Model | $H[P_{\text{train}}, \hat{P}]$ | $H[P_{\text{test}}, \hat{P}]$ | $n$-gram from train+test | | | | $n$-gram from test | | | |
|---|---|---|---|---|---|---|---|---|---|---|
| | | | $P_2$ | $P_3$ | $P_4$ | $P_5$ | $P_2$ | $P_3$ | $P_4$ | $P_5$ |
| 2×TRIL, K=10 | 11.56 | 12.27 | 10.4 | 12.8 | 20.9 | 30.7 | 13.1 | 21.9 | 49.6 | 81.1 |
| BASELINE-8D | 12.90 | 13.67 | 11.4 | 12.1 | 17.5 | 24.8 | 14.5 | 22.7 | 48.3 | 80.5 |
| ORACLE-TRAIN | – | – | 10.1 | 6.7 | 4.8 | 4.1 | 13.2 | 15.7 | 21.4 | 26.4 |
| ORACLE-TEST | – | – | 9.5 | 6.0 | 4.5 | 3.9 | 7.9 | 4.1 | 2.9 | 2.6 |

Table 2: Results for the alternative data split: Cross entropy and perplexity under $n = 2, 3, 4, 5$-gram language models estimated on either the full corpus or the test set only.

To give more insight into how the transition functions influence the results, Table 1a presents an exhaustive overview for all combinations of our simple flows. We observe that a powerful generative

flow is essential for successful models while the inference flow can remain relatively simple – yet simplistic choices, such as ID degrade performance. Choosing $F_g$ slightly more powerful than $F_q$ emerges as a successful pattern.

<table>
<tr><th rowspan="2"></th><th colspan="4">Flow $F_q$</th><th colspan="4">Flow $F_q$</th></tr>
<tr><th>ID</th><th>DIAG</th><th>TRIL</th><th>2×TRIL</th><th>ID</th><th>DIAG</th><th>TRIL</th><th>2×TRIL</th></tr>
<tr><td>ID</td><td>14.23±.00</td><td>14.23±.00</td><td>14.23±.00</td><td>–</td><td>0±.00</td><td>0±.00</td><td>0±.00</td><td>–</td></tr>
<tr><td>DIAG</td><td>12.82±.37</td><td>12.35±.37</td><td>12.20±.25</td><td>–</td><td>0.93±.15</td><td>0.85±.16</td><td>0.92±.13</td><td>–</td></tr>
<tr><td>TRIL</td><td>13.55±.01</td><td>11.99±.11</td><td>–</td><td>–</td><td>0.65±.01</td><td>0.95±.04</td><td>–</td><td>–</td></tr>
<tr><td>2×TRIL</td><td>–</td><td>11.86±.13</td><td>–</td><td>–</td><td>–</td><td>0.89±.07</td><td>–</td><td>–</td></tr>
</table>

(a) Test cross entropy $H[P_{\text{test}}, \hat{P}]$     (b) Average mutual information $\bar{I}$

Table 3: Results for different combinations of flows driving generative and inference transitions. A bar indicates combinations that did not allow for stable training. We also report ID for $F_g$ for completeness but note that it is by design unsiuted for this contextual setting.

## 5.4 Noise Model Analysis

We use $K = 20$ samples to approximate the entropy terms in (13). In addition we denote by $\bar{I}$ the average mutual information across all time-steps. Figure 3 shows how $\bar{I}$ along with the symbolic entropy $H[w_t|h_t]$ changes during training. Remember that in a non-autoregressive model, the latter corresponds to information that cannot be recovered in later timesteps. Over the course of the training, more and more information is driven by $\boldsymbol{\xi}_t$ and absorbed into states $\mathbf{h}_t$ where it can be stored.

Figures 1 and 1b show $\bar{I}$ for all trained models. In addition, Figure 3 shows a box-plot of $I(t)$ for each $t = 1 \ldots T$ for the configuration 2×TRIL, K=10. As initial tokens are more important to remember, it should not come as a surprise that $I(t)$ is largest first and decreases over time, yet with increased variance.

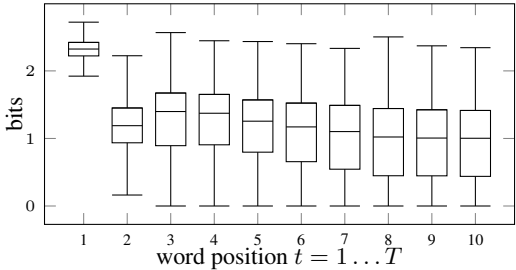

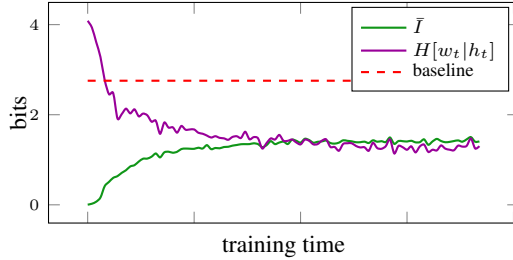

Figure 2: Noise mutual information $I(t)$ over sequence position $t = 1 \ldots T$.

Figure 3: Entropy analysis over training time. For reference the dashed line indicates the overall word entropy of the trained baseline.

## 6 Conclusion

In this paper we have shown how a deep state space model can be defined and trained with the help of variational flows. The recurrent mechanism is driven purely by a simple white noise process and does not require an autoregressive conditioning on previously generated symbols. In addition, we have shown how an importance-weighted conditioning mechanism integrated into the objective allows shifting stochastic complexity from the inference to the generative model. The result is a highly flexible framework for sequence generation with an extremely simple overall architecture, a measurable notion of latent information and no need for pre-training, annealing or auxiliary losses. We believe that pushing the boundaries of non-autoregressive modeling is key to understanding stochastic text generation and can open the door to related fields such as particle filtering [NLRB17, MLT+17].

## Footnotes

[1]For ease of exposition, we assume fixed length sequences, although in practice one works with end-of-sequence tokens and variable length sequences.

[2]Note that identifying $s$ as an invertible function allows us to perform a backwards density transformation which cancels the regularizing terms. This is akin to any flow objective (e.g. see equation (15) in[RM15]) where applying the transformation additionally to the prior cancels out the Jacobian term. We can think of $s$ as a stochastic bottleneck with the observation model $P(w_t|\mathbf{h}_t)$ attached to the middle layer. Removing the middle layer collapses the bottleneck and prohibits learning compression.

[3]Here we have tacitly assumed that $\mathbf{h}$ can be rewritten using the reprametrization trick so that the expectation can be expressed with respect to some parameter-free base-distribution. See [BGS15] for a detailed derivation of the gradients in (11).

[4]Note that for both data-space metrics there is a trivial generation system that achieves a 'perfect' score. Hence, both must be taken into account at the same time to judge performance.

[5]It should be noted that despite the greatly reduced vocabulary in character-level generation, RNN training without teacher-forcing for our data still fails miserably.

[6]Single best model with $d = 8$: $2 \times$ TRIL, $K = 10$ achieved $H[P_{\text{train}}, \hat{P}] = 11.26$ and $H[P_{\text{test}}, \hat{P}] = 11.28$.

[7]Note that the training-set oracle is not optimal for the test set. The entropy of the test set is 6.80.

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
