[Supplementary Material]

# A Detailed derivation of the weighted ELBO

We simplify the notation and write the distribution of the inference model over a subsequence $\mathbf{h}_{i...j}$ as $q(\mathbf{h}_{i...j}) = \prod_{t=i}^{j} q(\mathbf{h}_t|\mathbf{h}_{t-1}, w_{t...T})$ for any $1 \leq i \leq j \leq T$ without making the dependency on $\mathbf{h}_{i-1}$ and the data explicit. Furthermore, let $\mathcal{K}_t = \{\mathbf{h}_t^{(k)}\}_{k=1}^{K} \sim q(\mathbf{h}_t)$ be short for a set of $K$ samples of $\mathbf{h}_t$ from the inference model. Finally, let $\theta$ summarize all parameters of both, generative and inference model.

The key idea is to write the marginal as a nested expectation

$$P(\mathbf{w}) = \mathbb{E}_{q(\mathbf{h}_1)}\Big[P(w_1, \mathbf{h}_1)\mathbb{E}_{q(\mathbf{h}_{2...T})}\left[P(w_{2...T}, \mathbf{h}_{2...T}|\mathbf{h}_1)\right]\Big] \tag{14}$$

and observe that we can perform an MC estimate with respect to $\mathbf{h}_1$ only

$$P(\mathbf{w}) \approx \mathbb{E}_{\mathcal{K}_1}\Big[P(w_1, \mathbf{h}_1)\mathbb{E}_{q(\mathbf{h}_{2...T})}\left[P(w_{2...T}, \mathbf{h}_{2...T}|\mathbf{h}_1^{(k)})\right]\Big] \tag{15}$$

The same argument applies for $\frac{P(\mathbf{w},\mathbf{h})}{q(\mathbf{h})}$, the integrand in the ELBO. Now we can repeat the IWAE argument from [BGS15] for the outer expectation

$$\log P(\mathbf{w}) = \log \mathbb{E}_{q(\mathbf{h})}\left[\frac{P(\mathbf{w}, \mathbf{h})}{q(\mathbf{h})}\right] \tag{16}$$

$$= \log \mathbb{E}_{q(\mathbf{h}_1)}\left[\frac{P(w_1, \mathbf{h}_1)}{q(\mathbf{h}_1)}\mathbb{E}_{q(\mathbf{h}_{2...T})}\left[\frac{P(w_{2...T}, \mathbf{h}_{2...T}|\mathbf{h}_1)}{q(\mathbf{h}_{2...T})}\right]\right] \tag{17}$$

$$= \log \mathbb{E}_{\mathcal{K}_1}\left[\frac{1}{K}\sum_{k=1}^{K}\frac{P(w_1, \mathbf{h}_1^{(k)})}{q(\mathbf{h}_1^{(k)})}\mathbb{E}_{q(\mathbf{h}_{2...T})}\left[\frac{P(w_{2...T}, \mathbf{h}_{2...T}|\mathbf{h}_1^{(k)})}{q(\mathbf{h}_{2...T})}\right]\right] \tag{18}$$

$$\geq \mathbb{E}_{\mathcal{K}_1}\left[\log \frac{1}{K}\sum_{k=1}^{K}\frac{P(w_1, \mathbf{h}_1^{(k)})}{q(\mathbf{h}_1^{(k)})}\mathbb{E}_{q(\mathbf{h}_{2...T})}\left[\frac{P(w_{2...T}, \mathbf{h}_{2...T}|\mathbf{h}_1^{(k)})}{q(\mathbf{h}_{2...T})}\right]\right] = \mathcal{L} \tag{19}$$

$$\tag{20}$$

where we have used the above factorization in (17), MC sampling in (18) and Jensen's inequality in (19). Now we can identify

$$\omega_1^{(k)} = \frac{P(w_1, \mathbf{h}_1^{(k)})}{q(\mathbf{h}_1^{(k)})}\mathbb{E}_{q(\mathbf{h}_{2...T})}\left[\frac{P(w_{2...T}, \mathbf{h}_{2...T}|\mathbf{h}_1^{(k)})}{q(\mathbf{h}_{2...T})}\right] \tag{21}$$

and use the log-derivative trick to derive gradients

$$\nabla\mathcal{L} = \mathbb{E}_{\mathcal{K}_1}\left[\sum_{k=1}^{K}\frac{\omega_1^{(k)}}{\sum_{k'}\omega_1^{(k')}}\nabla\log\omega_1^{(k)}\right] \tag{22}$$

Again, we have omitted carrying out the re-parametrization trick explicitly when moving the gradient into the expectation and refer to the original paper for a more rigorous version. The gradient of the logarithm decomposes into two terms,

$$g_t^1 = \nabla\log\frac{P(w_1, \mathbf{h}_1)}{q(\mathbf{h}_1)} \tag{23}$$

$$g_t^2 = \nabla\log\mathbb{E}_{q(\mathbf{h}_{2...T})}\left[\frac{P(w_{2...T}, \mathbf{h}_{2...T}|\mathbf{h}_1)}{q(\mathbf{h}_{2...T})}\right] \tag{24}$$

The first is the contribution to our original ELBO normalized by the IWAE MC weights. The second is identical to our starting-point in (16) but for $t = 2 \ldots T$ and conditioned on $\mathbf{h}_1^{(k)}$. Iterating the above for $t = 2 \ldots T$ yields the desired bound.

To allow tractable gradient computation using the importance-weighted bound, we use two simplifications. First, we limit the computation of the weights $\omega_t^{(k)}$ to a finite horizon of size 1 which reduces them to only the first factor in (21). Second, we forward only a single sample $\mathbf{h}_t$ to the next time-step to remain in the usual single-sample sequential ELBO regime (which is important as $g_t^2$ depends on $\mathbf{h}_{t-1}$). That is, we sample $\mathbf{h}_t$ proportional to the weights $\omega_t^{(k)} \ldots \omega_t^{(k)}$. A more sophisticated solution would be to incorporate techniques from particle filtering which maintain a fixed-size sample population $\{\mathbf{h}_t^{(1)}, \ldots, \mathbf{h}_t^{(K)}\}$ that is updated over time.