[Reviews · NeurIPS 2018]

Reviewer 1



This paper proposes a non-autoregressive state space model for sequence generation. It builds on flow-based variational inference and uses an importance weighted evidence lower bound to shift the complexity from the generative to the inference model. The paper then performs evaluation on word generation from the BooksCorpus dataset. Quality: Strengths: The paper is well-motivated and tackles a common problem: non-autoregressive generation of sequences. The model is elegant and the results are thorough and strong compared to the autoregressive baseline. The graphs of the entropy gap and entropy analysis are useful. Weaknesses: The limit in word length to be at most length 12 makes it hard to know if this approach generalizes to longer sequences. For example, at what sequence lengths does this start to break down? What is the cross-entropy of the baseline that was trained without autoregressive inputs? The network sizes are quite small. Only state sizes of 8 and 16 are considered which are much smaller than standard models. Why weren't larger state sizes used? Table 2 seems like it should also include results from Real-NVP. Clarity: Weaknesses: The description of normalizing flows in the last part of section 2.2 could be much improved. Lin 27: 'which may constraint' -> which may constrain Line 34: Typo: prevoiusly Line 46: determinstic Footnote 2: Jaccobian Algorithm 1: Details -> Detailed Line 212: emphasise -> emphasis Originality: The method is relatively original and the coupled transition functions are quite interesting and are novel compared to existing work. Significance: The work has significance for the growing area of non-autoregressive sequence generation. === POST-REBUTTAL === Thanks for responding to my questions. I have kept my accept score.

Reviewer 2



This paper introduces a probabilistic model for unconditional word generation that uses state space models whose distributions are parameterized with deep neural networks. Normalizing flows are used to define flexible distributions both in the generative model and in the inference network. To improve inference the inference networks uses samples from the prior SSM transitions borrowing ideas from importance-weighted autoencoders. I enjoyed reading this paper, as it gives many useful insights on deep state space models and more in general on probabilistic models for sequential data. Also, it introduces novel ways of parameterizing the inference network by constructing a variational approximation over the noise term rather than the state. I appreciated the usage of unconditional word generation as a simple example whose results are easier to interpret, but the validation of these results to more complex sequential tasks would make this a much stronger submission: - I agree on the discussions on the importance of not having the autoregressive component as most of the models in the literature. In this paper however you only focus on the "simple" sequential task of word generation, while in many cases (e.g. audio or video modelling) you have very long sequences and possibly very high dimensional observations. The natural question is then how do these results generalize to more complex tasks? Would you for example still be able to train a model with a higher dimensional state space without the autoregressive component, performing annealing or having auxiliary losses? - In more complex tasks you often need some memory in the model (e.g. LSTM units) which is why previous works on deep state space models combine RNNs with SSMs. How would you add memory to your model? Would your conclusions still hold in this case? Also, more in depth analysis of the novel components of your model would have been useful: - When defining the importance weighted ELBO, you use many different samples from the generative model as an input to the inference network. Previous work used for example just the prior mean in the inference network, instead of taking all the samples which is computationally intensive. How does the model perform if for example you only pass the prior mean instead of all the samples? - What happens if you only use normalizing flows in generative model/inference networks, and not in both? This is more commonly done in previous papers. Minor comments - The results on the set-based metrics and the ones in table 2 should be discussed more in depth - Is your citation style in line with the nips guidelines? Usually nips paper have numbered citations Typos: Line 6: built Line 281: samples

Reviewer 3



Summary This paper proposes a state space model for sequence generation, where the generative model is non-autoregressive but the (variational) inference network can condition state predictions on the observed sequence. The model is evaluated on character-based lexicon generation (generating words independent of any context). Quality The paper proposes a novel, principled method for learning a non-autoregressive sequence model. I am not familiar enough with flow-based VAE to verify the model derivation, but it seems reasonable. My main concerns are with the experimental setup: Generating words in isolation is a relatively restricted task, and I don't see why the model should not be applied to (word-level) sentence generation, even if at first that mainly points out limitations of the model. The train/test split in which the test set words are completely disjoint from the training words makes more sense than the one where the same words can appear in the train and test data, so I would like to see more of a comparison between the different models there. Clarity Section 2 focuses on how the model is trained, but I would like to see a more algorithmic explanations of how the method is applied to sequence generation, for either pure sampling or inference given an observed sequence. In Algorithm 1 there are two hidden state values, h^k_g and h^k - it is not clear why, or how q_k() is defined. Significance This is an interesting model, and the proposed model obtains lower cross-entropy than a comparable baseline. I am inclined to recommend acceptance. However it is not entirely clear to me what the proposed model's advantages are, or what the results tell us about the problem being studied, so I would like to see clearer motivations and analysis. It seems that for unconstrained sampling from the model it is purely autoregressive, but during inference given an observed sequence the inferred noise variable is supposed to capture part of the variance which could otherwise have been captured by an auto-regressive model. Does this mean the we need, or don't need, the autoregressive property for the problem studied? --- Thanks for clarifying the motivation.